# Effect of Lean Red Meat from Beef (Pirenaica Breed) Versus Lean White Meat Consumption on Diet Quality: A Randomized-Controlled Crossover Study in Healthy Young Adults

**DOI:** 10.3390/nu15010013

**Published:** 2022-12-20

**Authors:** Maria Luisa Miguel-Berges, Marta Fajó-Pascual, Luis A. Moreno, Marimar Campo, Ana Guerrero, Jose Luis Olleta, Pilar Santolaria Blasco, Alba M. Santaliestra-Pasías

**Affiliations:** 1GENUD (Growth, Exercise, Nutrition and Development) Research Group, Facultad de Ciencias de la Salud, Universidad de Zaragoza, 50009 Zaragoza, Spain; 2Instituto Agroalimentario de Aragón (IA2), 50013 Zaragoza, Spain; 3Instituto de Investigación Sanitaria Aragón (IIS Aragón), 50009 Zaragoza, Spain; 4Centro de Investigación Biomédica en Red de Fisiopatología de la Obesidad y Nutrición (CIBERObn), Instituto de Salud Carlos III, 28029 Madrid, Spain; 5Departamento de Producción Animal y Ciencia de los Alimentos, Universidad de Zaragoza-CITA, C. de Miguel Servet 177, 50013 Zaragoza, Spain; 6Departamento de Producción Animal y Ciencia de los Alimentos, Escuela Politécnica Superior de Huesca, Instituto Universitario Ciencias Ambientales, Universidad de Zaragoza, Ctra. Cuarte, 22004 Huesca, Spain

**Keywords:** beef (Pirenaica breed), young adults, Spanish study, crossover trial design

## Abstract

A randomized crossover study was carried out in three University accommodation halls. Participants consumed either beef (Pirenaica breed) (PB) or conventional chicken (CC) three times per week for an 8-week periods with their usual diet, each one separated by a 5-week wash out period. Dietary variables were recollected by the Food Frequency Questionnaire (FFQ), and the Diet Quality Index (DQI) was calculated. Forty-seven healthy adults were included (19.9 ± 1.75 years). The inclusion of both types of diets did not modify the components of the DQI, such as the diversity, equilibrium, adequacy and excess. However, when only the first period was analyzed, a significant decrease in the consumption of fruits and vegetables was observed in those participants who received the PB diet (intervention group). The CC diet (control group) significantly reduced the consumption of fish and eggs, total DQI, and DQI quality component. The expected effect was observed in the significant increment of consumption of red meat after the intervention period.

## 1. Introduction

The consumption of red meat has been discussed in the recent decade to the point that red meat consumption was classified as probably carcinogenic to humans (Group 2^a^); the Working Group defined red meat as unprocessed mammalian muscle meats, including beef, pork, lamb, mutton, horse, or goat meat [1]. The association with cancer seems clear for the consumption of processed meats, such as sausages, burgers etc.; however, the findings are inconsistent for the consumption of non-processed red meat [2]. Moreover, according to the 2015 US Dietary Guidelines (USDA), red meat is an exceptional source of iron, an essential mineral responsible for the transport of oxygen in the blood, whose deficiency can lead to anemia. These guidelines suggest that healthier ways to include red meat in our diets are needed, such as excluding the fatty part of meat [3]. In Spain, the qualitative recommendation is that not necessarily individuals should consume meat every day; in fact, it is advisable to alternate it with fish and to include different meat from different species such as beef, pork, chicken, rabbit, lamb, etc. [4].

In recent years, the assessment of the overall diet quality through scoring systems has been extensively investigated, and increasingly applied in epidemiological studies. This concept involves the assessment of both quality, variety and equilibrium of the whole diet, enabling examination of associations between foods and/or food groups and health status, rather than just nutrients [5]. Diet quality is measured by scoring food patterns in terms of how closely they align with national dietary guidelines and how diverse the variety of healthy choices is within core food groups [6]. Therefore, it is important to assess the quality of the overall diet since it has been shown that, regardless of which index is used for diet quality assessment, adherence to a healthy diet has been associated with a reduced risk of chronic diseases [7].

In Europe, public administrations have recently encouraged extensive livestock farming systems, usually related to a high quality meat [8]. Moreover, in order to promote the binomial meat and sustainability, it is important to evaluate alternatives such as increasing the demand for locally produced beef in order to reduce the carbon footprint and reduce the price of the product [9]. For this reason, bovine meat produced from animals born, raised on local feedstuffs, slaughtered, processed and even consumed in specific geographical areas such as the Pyrenees, might be a more sustainable livestock production alternative [10]. The Pyrenees (mountains located in the border between France and Spain) are an optimal place for the practice of extensive livestock rearing given the richness of forages and the existence of native beef breeds that make use of these natural feedstuffs. In these mountains, young calves of native breeds are raised mainly on local pastures with a recognized beneficial impact in environmental and social sustainability of this rural geographical area.

The aim of the present study is to assess the effect of the consumption of lean red meat from beef- (Pirenaica breed) (PB) versus lean white meat (conventional chicken) (CC) on specific food groups and the diet quality index and in institutionalized young adults from Aragon, Spain.

## 2. Materials and Methods

### 2.1. Sample

Assuming a two-tailed alpha error of 0.05, with a statistical power of 90%, and dropout rates of up 20%, the required sample size was 60 participants in total, 30 per group. Study participants were randomized 1:1 into two equally sized groups. Computer generated random allocation was centrally elaborated, and stratified by sex. The procedure was internet-based and developed by one investigator of the team (MLMB). Another investigator was the responsible to enrolled participants (AMSP). However, and despite all the efforts, of the 52 students who agreed to participate in the study, 2 dropped out before starting intervention period 1 measurements, and 3 dropped out before starting intervention period 2. Finally, a total of 47 participants (24 males, 51.1%) were recruited whose age ranged between 18 to 27 years. Three University students’ halls of residence, two of them in the city of Huesca and one in the city of Zaragoza (Spain), were selected. A study information sheet on the nature and purpose of the study was given to all participants, and a written informed consent was obtained. After that, they were considered for inclusion in the study. Eligibility criteria were free of any chronic, metabolic, endocrine or nutrition-related disease. No participant reported to be currently enrolled in a weight loss program, or currently taking any medications known to have a lipid-lowering effect. Authorization from the Research Ethics Committee of the Government of Aragon (Spain) was obtained (N° 17/2018). The study was performed following the ethical guidelines of the Edinburgh revision of the 1964 Declaration of Helsinki (2000).

### 2.2. Experimental Design

The current study is a sub-study included in the DIETAPYR2 project (Innovaciones aplicadas a la cadena productiva pirenaica de vacuno para valorizar una carne identificable por el consumidor), a randomized-controlled crossover trial (NCT 04832217) consisting of two experimental periods with a duration of 8 weeks each. Enrolled participants were randomly assigned either to a lean red meat from beef- (Pirenaica breed) diet or to a lean white meat diet (Conventional chicken) and were instructed to consume the allocated type of meat, three times per week. To ensure harmonization in the study, each chef of the 3 selected student accommodation halls was provided with instructions on the cooking methods for chicken and beef and they also received examples of recipes. Meat was breaded, stewed and grilled for beef and breaded, stewed and roasted for chicken.

During the 2 weeks before the first intervention period, researchers contacted with participants and obtained their informed consent. Then, the first visit was scheduled to fill the socio-demographic questionnaire and food frequency questionnaire (Appendix A). At the time of this visit, each participant was randomly assigned to PB diet (intervention group) or CC diet (control group).

Following an 8-week experimental period, participants were invited to fill the same questionnaires as during the first visit. Afterwards, a 5-week washout period took place to remove the possible residual effects of the first experimental period. Participants were requested not to change their diet or physical activity habits for the 5-week washout period. After the washout period, participants were asked again to perform the same questionnaires during a 3rd visit right before the second experimental period (period 2). During this period those previously on the PB diet (intervention group) were crossed to the CC diet (control group) and vice versa. Finally, after the 8-week second experimental period, participants were assessed again as in the previous visits.

### 2.3. Dietary Assessment

To assess the consumption of foods and beverages, participants were asked to complete food frequency questionnaires (FFQ) right before and after the two 8-week experimental periods. The questionnaire was based on a previously validated questionnaire [11,12]. The FFQ consisted of 133 items with an indication of serving size, organized by food groups. The categories of response were never or almost never; 1–3 per month; 1 per week; 2–4 per week; 5–6 per week; 1 per day; 2–3 per day and 4–6 per day. Dietary data from the FFQ were converted to average daily intake values (e.g., 1 serving/week = 0.14 serving/d). A specific question about the number of days per week fruit was consumed as dessert was also included in the questionnaire. Moreover, for current analysis, 8 food groups were chosen: (1) fruits and vegetables, (2) dairy products, (3) sweetened beverages, (4) sweets, (5) desserts, (6) fish and eggs, (7) white meat and (8) red meat.

After collecting the food consumption data from the FFQ, the Diet Quality Index (DQI) was calculated. The DQI is based on the DQI validated by Huybretchs et al. [13] for preschoolers, and adapted by Vyncke et al. [14] later for adolescents who assess the degree of adherence to the Food Based Dietary Recommendations (RDBAs) of the Health Authority of the Flemish community of Belgium. The DQI is an index largely used in studies to assess the overall quality of the diet. This index consists of three components: dietary quality, dietary diversity and dietary equilibrium. Dietary quality expressed whether the children individual made the optimal food quality choices within a certain food group represented by foods from a “preference” group (e.g., fresh fruit), followed by an “intermediate” group (e.g., white bread), and a ‘low-nutrient, energy-dense group’ (e.g., sweet drinks). Dietary diversity expressed the degree of variation in the diet. This diversity component was obtained by giving points ranging from 0 to 9 when at least one serving of food of a recommended food group was consumed. For this, the recommended food groups were: (1) water (drinks); (2) bread and cereals; (3) grains and potatoes; (4) vegetables; (5) fruit; (6) dairy products; (7) cheese; (8) meat, fish, eggs and substitutes and (9) fats and oils. In addition, a last group was considered compounded of non-recommended foods i.e., sporadic consumption. Dietary equilibrium was calculated from the difference between the adequacy of the diet (the percentage of the minimum recommended intake for each of the main food groups truncated to 1) and the excess of the intake in the diet (the percentage of intake exceeding the upper level of the recommendation, truncated to 1 if >1 and truncated to 0 when < 0). These three components of the DQI were expressed in percentages. The dietary quality component ranged from −100 to 100%, while dietary diversity and dietary equilibrium ranged from 0 to 100%. The DQI was calculated as the mean of the three components and as such, its value ranged from a minimum of −33 to a maximum of 100% with a higher score representing a better adherence to the dietary guidelines of the participant. Negative values are possible but would represent situations where the participant mostly consumed foods from the consumption group occasional (low in nutrients and high in energy).

### 2.4. Statistical Analysis

Means and standard deviations were used to describe the magnitude and variability of the continuous variables showing a Gaussian distribution. The validity of the crossover design (i.e., the absence of a carryover and period effect) was tested by a two-factor repeated measures analysis of variance (rmANOVA) model, where the diet sequence (Pirenaica breed_Conventional Chicken) versus (Conventional Chicken_Pirenaica breed) was the between-participants factor and the type of diet (PB versus CC) was the within-participants factor. When no statistical significance was found for the diet sequence factor, diet outcome comparisons were made through the parametric t-test for paired samples. For variables where the diet sequence factor was statistically significant i.e., some food groups (fish and eggs, fruits and vegetables, dairy products, sweetened beverages, sweets and white meat) DQI score and DQI score quality component, only the first period data were analyzed through a two-factor rmANOVA, adjusted for sex and age; the diet was the between-participants factor and the time before and after the period was the within-participants factor. The difference between the end and the beginning of each period by diet was further analyzed with a paired *t*-test. All analyses were done using Statistical Package for the Social Sciences (SPSS Version 21 for Windows; SPSS Inc., Chicago, IL, USA). Statistical tests were two-tailed, and findings were considered statistically significant at *p* < 0.05.

## 3. Results

Table 1 shows the characteristics of the study sample (*n* = 47) at the onset of the study being 51.1% male. The mean age of the participants was 19.90 (± 1.75) years. Approximately 25% and 26% of the participants’ mothers and fathers, respectively, were categorized in the higher educational level. Significant differences were found between the CC control and the PB intervention diets for gender, and age. The male’s proportion and the age of the participants was significantly higher in the PB diet group.

The differences in some food groups consumption, DQI quality component and total DQI mean scores by diet during the first experimental period after gender and age adjustment are shown in Table 2. The data are shown only for this period since a diet sequence effect was statistically significant both for some food groups, the DQI quality component and the total DQI. A significant decrease in the fish and eggs consumption (Mean −54.81; CI: −98.08; −11.53), in the total DQI (Mean −3.83; CI: −6.74; −0.93), as well as in the quality component (Mean −9.50; CI: −18.70; −0.30) was observed in those participants who received the CC diet (control group) while the decrease was nonsignificant for those on the PB diet (intervention group). However, this different food consumption of fish and eggs (F = 2.67 *p* = 0.109), for the total DQI (F = 1.47 *p* = 0.232), and DQI quality component (F = 1.22 *p* = 0.276) were nonsignificant between both diets. On the other hand, a significant decrease in the fruits and vegetables’ consumption (Mean −205.78; CI: −392.68; −18.88) was observed in those participants who received the PB diet (intervention group) during the first period. However, this difference was nonsignificant between diets (F = 1.51 *p* = 0.226).

Table 3 shows the results of the diet outcome comparisons for those variables which did not show a sequence effect through a 2-factor rmANOVA analysis and hence a crossover design analysis was undertaken in the total sample through a parametric t-test for paired samples. A small nonsignificant decrease in the frequency of weekly fruit consumption as dessert, as well as desserts consumption and in the DQI excess component were observed for the PB diet compared to the CC diet. On the other hand, as expected, it was observed a significant increase in red meat consumption for the PB diet (intervention) compared to the CC diet (control). Additionally, a small, nonsignificant increase in the DQI diversity, equilibrium and adequacy components were observed for the PB diet (intervention) compared to the CC diet (control).

## 4. Discussion

In this study, the possible effect of the consumption of beef (Pirenaica breed) diet versus conventional chicken diet on specific food groups, the weekly frequency of fruit as dessert, DQI components and the total DQI has been examined. Our results show that an intensive dietary intervention with inclusion of PB or CC meat for three-days weekly did not significantly modify the weekly frequency of fruit consumption as dessert neither the components of DQI such as diversity, equilibrium, adequacy and excess, but as expected, red meat consumption increased after the intervention period with beef (Pirenaica breed). However, when only the first period was analyzed since a diet sequence effect was detected, a significant decrease in the consumption of fish and eggs and in the score of quality component and total DQI were observed in those participants who received the CC diet (control group). On the other hand, a significant decrease in the consumption of fruits and vegetables was observed in those participants who received the PB diet (intervention group). The CC diet (control group) significantly reduced the consumption on fish and eggs. On the other hand, the consumption of fruits and vegetables was significantly reduced in those participants who consumed PB diet (intervention group). The expected effect was observed in the significant increment of consumption of red meat after the intervention period.

Both diets (CC and PB) reduced the quality component of DQI and total DQI, but only the CC diet (control group) reduced these values significantly. The diversity component improved in those participants consumed PB diet, but it was not significant.

In general, the diet quality of participants in the study is relatively low (<50), but, in the same line, recent studies point out that the majority of the population needs changes in their feeding patterns [15]. Previous studies have shown that the DQI is positively associated with nutrient intake adequacy [16,17]. In a recent meta-analysis, it was observed than higher DQI scores were associated with a significant reduction in the risk of all-cause mortality (22%), cardiovascular (22%), cancer (16%), type 2 diabetes (18%), and neurodegenerative diseases (15%) [18]. A cross-sectional study carried out in the United States in adolescents and young adults concluded that diet quality was low and associated with a high odds of hypertension [19].

To the authors knowledge, no studies were published which assess the total DQI or the DQI components and their relationship with an intervention aiming to increase the intake of specific food products associated with a rural territory. Foods that provide a mode of subsistence of its inhabitants while making use of local natural resources in an environment respectful manner as it is the case for beef (Pirenaica breed). This kind of meat originated from autochthonous Spanish breeds reared in extensive production systems and whose conservation support rural development avoiding rural depopulation [20]. Some studies suggest that meat from calves reared and fed mostly on local feedstuffs, had a proportion of saturated versus unsaturated fat, as well as between omega-3 fatty acids versus omega-6, within the recommended levels for developing a healthy diet [8].

The evidence of diseases related to the consumption of red meat, and therefore, dietary recommendations to limit red meat are based on observational studies linking intake to cardiovascular disease risk due to its content on saturated fatty acids [21]. However, the results of our study are in line with other studies comparing diets based on red and white meat, based on their lipid and lipoprotein effects, which did not provide evidence for choosing white over red meat for reducing cardiovascular disease risk, i.e., moderate consumption of red meat did not seem to worsen the lipid profile of consumers [22]. Likewise, Santaliestra-Pasias et al., after analyzing the body composition indicators, lipid profile and fatty acids of the same participants of this study concluded that following a conventional chicken or a beef (Pirenaica breed) diet did not modify the lipid and lipoprotein profile in institutionalized young health participants (submitted results). These results are consistent with some previous results of intervention studies, focusing on the comparison of lean lamb and chicken diets in institutionalized populations. Both, in young adults [23], and older women [24], the results showed no effect on the main body composition or cardiovascular diseases indicators. Moreover, the young adults study reported some additional beneficial effects in the intervention group, for example, the triglycerides concentration decreased after 8 weeks of lean lamb-based diet consumption. Other intervention study whose objective were assess the effect of beef and chicken consumption on plasma lipid levels in hypercholesterolemic men could conclude that comparably lean beef and chicken had similar effects on plasma levels of total, low-density lipoprotein, high-density lipoprotein cholesterol and triglycerides concluding that lean beef and chicken are interchangeable in a diet [25]. In this same line, Davidson et al. [26] after evaluating two types of diet compounded by beef, veal, and pork versus poultry and fish, did not found differences between diets in the concentrations of total cholesterol and low-density lipoprotein cholesterol being nearly identical between treatments.

### Strengths and Limitations

The study design was a randomized crossover trial, which allows comparing the obtained diets outcomes within the participants that acted as their own controls, reducing the expected variability. Hence compared with a parallel group design, fewer participants are needed to obtain the same power [27]. The inclusion of an institutionalized young adult population provided a sample with similar dietary and lifestyle patterns, reinforcing the validity of the results.

To assess the dietary intake, a self-reported FFQ was used due to its practicality but more objective dietary assessment methods are required; however, they more laborious, such as dietary records, allowing to measure nutrient intake with precision [5]. Additionally, we could not rule out the carryover, and period effect in all the variables, and some of them could only be analyzed with data from the first period of the study, reducing the sample size. Moreover, sample size was similar, but it might not suffice to detect changes in diet quality index.

## 5. Conclusions

In our study, the effect of conventional chicken versus beef (Pirenaica breed) diet in the diet quality index and their components in young people were assessed. The inclusion of both types of diets did not modify the components of the diet quality index such as the diversity, equilibrium, adequacy and excess, nor the quantity of the days the participants chose fruit as dessert. However, when only the first period was analyzed for total DQI and DQI quality component, a significant decrease was observed in those participants who received the conventional chicken, and no effect was observed in the PB diet. These results made us realize that a type of meat resulting from an extensive meat production system, which guarantees animal welfare, and good production practices, could be incorporated into the dietary guidelines as a part of a healthy and sustainable diet.

## Figures and Tables

**Table 1 nutrients-15-00013-t001:** Sociodemographic characteristics of included participants.

		Total *n* = 47	Participants Who Started with the Beef (Pirenaica Breed) Diet *n* = 24	Participants Who Started with the Conventional Chicken Diet *n* = 23	*p*
Gender (*n*, %)	Female	23 (48.9)	7 (29.2)	16 (69.6)	0.006
	Male	24 (51.1)	17 (70.8)	7 (30.4)	
Age (mean years ± SD)		19.90 ± 1.75	20.49 ± 2.09	19.29 ± 1.02	0.022
Maternal education, *n* (%)	Low	10 (21.3)	4 (16.7)	6 (26.1)	0.691
	Medium	25 (53.2)	14 (58.3)	11 (47.8)	
	High	12 (25.5)	6 (25)	6 (26.1)	
Paternal education (*n*, %)	Low	13 (28.9)	6 (25)	7 (33.3)	0.814
	Medium	20 (44.4)	11 (45.8)	9 (42.9)	
	High	12 (26.7)	7 (29.2)	5 (23.8)	

Significant differences (*p* < 0.05).

**Table 2 nutrients-15-00013-t002:** Differences in some food groups, DQI quality component and total DQI after Conventional Chicken versus beef (Pirenaica breed) diets during period ^1^.

	Conventional Chicken (CC) Diet *n* = 23					Beef (Pirenaica Breed) Diet (PB) *n* = 24						
	Before	After	Mean Change	95% CI	*p*	Before	After	Mean Change	95% CI	*p*	Difference in Mean Change between Diets (CC-PB Diets) ^1^	*F* (*p*)
	Mean ± SD	Mean ± SD				Mean ± SD	Mean ± SD					
Fish and eggs ^2^	140.28 ± 18.29	85.48 ± 14.01	−54.81	−98.08; −11.53	0.016	138.50 ± 17.67	137.11 ± 16.49	−1.39	−35.28; 32.49	0.933	−53.42	2.67 (0.109)
Fruits and Vegetables ^2^	622.23 ± 69.62	511.36 ± 54.53	−110.88	−245.16; 23.40	0.101	797.85 ± 113.10	592.06 ± 79.19	−205.78	−392.68; −18.88	0.032	94.9	1.51 (0.226)
Dairy products ^2^	271.31 ± 33.66	309.21 ± 57.24	35.91	−63.63; 135.45	0.461	357.84 ± 56.73	279.46 ± 46.63	−78.37	−184.62; 27.87	0.140	114.28	2.64 (0.111)
Sweetened beverages ^2^	197.35 ± 58.39	138.16 ± 21.78	−59.18	−159.86; 41.49	0.235	235.50; 67.48	151.02 ± 53.63	−84.47	−172.44; 3.05	0.059	25.29	1.89 (0.184)
Sweets ^2^	69.31 ± 13.77	56.86 ± 10.24	−12.44	−35.61; 10.72	0.276	86.79 ± 19.75	93.63 ± 17.72	6.56	−32.38; 45.52	0.730	−19	0.28 (0.599)
White meat ^2^	112.57 ± 13.76	101.15 ± 9.95	−11.41	−41.06; 18.22	0.432	103.04 ± 10.75	87.26 ± 10.11	−15.78	−44.15; 12.58	0.261	4.37	0.06 (0.806)
DQI quality	32.914 ± 4.8	23.414 ± 4.8	−9.50	−18.70; −0.30	0.044	31.933 ± 4.40	29.089 ± 6.00	−2.84	−12.97; 7.28	0.566	−6.66	1.22 (0.276)
Total DQI	45.12 ± 4.69	41.28 ± 4.69	−3.83	−6.74; −0.93	0.012	44.51 ± 4.71	43.22 ± 4.75	−1.29	−4.63; 2.06	0.433	−2.54	1.47 (0.232)

Adjusted by gender and age. ^1^ The change over time is different for each diet; ^2^ g/day, DQI: dietary quality index (score), Significant differences (*p* < 0.05).

**Table 3 nutrients-15-00013-t003:** Differences in some food groups, DQI diversity and equilibrium components and fruit consumption as dessert, after Conventional Chicken versus beef (Pirenaica breed) diets in both periods analyzed through a crossover design analysis.

	After Conventional Chicken (CC) Diet *n* = 47	After Beef (Pirenaica Breed) (PB) Diet *n* = 47	Difference in Mean Change between Diets (CC-PB Diets)		
	Mean ± SD	Mean ± SD	Mean Change ^1^	95% CI	*F* (*p*)
DQI diversity	95.50 ± 0.94	96.71 ± 0.82	−1.21	−0.325; 2.740	2.52 (0.119)
DQI equilibrium	4.44 ± 0.17	4.48 ± 0.15	−0.04	−0.317; 0.382	0.038 (0.846)
Adequacy	55.57 ± 1.97	55.72 ± 1.94	−0.15	−3.274; 3.585	0.443 (0.509)
Excess	23.22 ± 1.61	23.05 ± 1.28	0.17	−2.285; 1.946	1.383 (0.246)
Days/weekly fruit as dessert	5.10 ± 0.27	4.98 ± 0.25	0.13	−0.536; 0.282	0.039 (0.845)
Desserts ^2^	40.60 ± 7.11	35.92 ± 6.01	4.68	−22.68; 13.53	0.268 (0.607)
Red meat ^2^	22.79 ± 2.86	54.52 ± 3.75	−31.73	22.64; 40.81	49.53 (0.000)

Significant differences (*p* < 0.05), DQI: dietary quality index, ^1^ Mean change: CC-PB diets, ^2^ g/day.

## Data Availability

Available information about the study will be included at http://dietapyr2.com (accessed on 6 September 2022).

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
