# Peer review of "Effect of Lean Red Meat from Beef (Pirenaica Breed) Versus Lean White Meat Consumption on Diet Quality: A Randomized-Controlled Crossover Study in Healthy Young Adults"

_nutrients, 2022, doi:10.3390/nu15010013_

Round 1

Reviewer 1 Report

I would like to thank the authors for providing important information about the effect of the consumption of red beef meat from a local breed on specific food groups and the diet quality index.

The effect of red meat consumption on human health is a very “hot” topic with controversial results in the literature. Thus, any relevant information can be useful for the scientific community, the meat industry, but also for the public.

The present manuscript is well written, well structured and adequately cited.

Tables 1-3 are all in different format, which makes it difficult to follow, even for experienced readers and especially Table 2.

In the “Experimental design” section of the study a lot of samples were taken from the participants but there is no mention in the results session. There is also no clear presentation of results of these tests of the participants after the diet intervention.

Unfortunately I can not suggest the publication of the paper in its current form since critical information is missing.

Author Response

Dear Editor,

We would like to submit a revised version of our manuscript " Effect of lean red meat from beef- (Pirenaica breed) versus lean white meat consumption on diet quality: a randomized-controlled crossover study in healthy young adults". We would like to thank the possibility to review again the document and we consider that the manuscript has included the reviewer’ suggestions in the new version. Changes in the original manuscript are marked up using the “Track Changes” function.

Reviewer 1:

I would like to thank the authors for providing important information about the effect of the consumption of red beef meat from a local breed on specific food groups and the diet quality index. The effect of red meat consumption on human health is a very “hot” topic with controversial results in the literature. Thus, any relevant information can be useful for the scientific community, the meat industry, but also for the public. The present manuscript is well written, well structured and adequately cited.

Comment 1: Tables 1-3 are all in different format, which makes it difficult to follow, even for experienced readers and especially Table 2.

Answer: Thank you for your valuable comments. We agree with you and the tables have been modified accordingly. We believe that they are now easier to follow.

Comment 2: In the “Experimental design” section of the study a lot of samples were taken from the participants but there is no mention in the results session. There is also no clear presentation of results of these tests of the participants after the diet intervention.

Answer: We appreciate your comment. Our article is the continuation of an article published in this journal under the title "Effect of the intake of lean red-meat from beef- (Pirenaica breed) versus lean white-meat on body composition, fatty acids profile and cardiovascular risk indicators: a randomised cross-over study in healthy young adults"[1] where other variables were analysed. For this reason, the authors have based on your comment, to eliminate, into the methodology section, those variables that have not been analysed in this article.

  1. Santaliestra-Pasías, A.; Miguel-Berges, M.; Campo, M.; Guerrero, A.; Olleta, J.; Santolaria, P.; Moreno, L. Effect of the Intake of Lean Red-Meat from Beef-(Pirenaica Breed) versus Lean White-Meat on Body Composition, Fatty Acids Profile and Cardiovascular Risk Indicators: A Randomized Cross-Over Study in Healthy Young Adults. Nutrients 2022, 14, doi:10.3390/nu14183724.

Reviewer 2 Report

This study assessed the effect of a traditional chicken and beef (Pirenaica breed) diet on the diet quality index and their components in young adults.

Since this study emphasizes that the students surveyed are randomly invited, the effective sample size is only 47, and they are from 3 different universities. In this case, is the research data sufficiently representative? Supplementary explanation is suggested.

Is there any data comparison between different universities in this study? If yes, please explain whether the trend is consistent?

This study should have the analysis data in gender differences. However, the effective sample size is only 47. Is it impossible to conduct gender analysis? Supplementary explanation is suggested.

In order for readers to have a better understanding of the content of the food frequency questionnaires (FFQ) in this study, it is recommended that the complete information of the FFQ be showed as appendix.

Table 1 and Table 2 are very messy, and there are many abnormal word breaks, which are difficult to read, please adjust.

Citations listed in this manuscript do not conform to the format of this journal. In addition, links such as DOI or [Google Scholar] [CrossRef] [PubMed] of each article are not listed. Please correct and add relevant information.

Author Response

Dear Editor,

We would like to submit a revised version of our manuscript " Effect of lean red meat from beef- (Pirenaica breed) versus lean white meat consumption on diet quality: a randomized-controlled crossover study in healthy young adults". We would like to thank the possibility to review again the document and we consider that the manuscript has included the reviewer’ suggestions in the new version. Changes in the original manuscript are marked up using the “Track Changes” function.

Reviewer 2:

This study assessed the effect of a traditional chicken and beef (Pirenaica breed) diet on the diet quality index and their components in young adults.

Comment 1. Since this study emphasizes that the students surveyed are randomly invited, the effective sample size is only 47, and they are from 3 different universities. In this case, is the research data sufficiently representative? Supplementary explanation is suggested.

Answer: Thank you for your comment. The students are from different cities of Spain but  during the academic year they live in Zaragoza and Huesca at 3 different student’s residences. We don’t consider that the sample is representative but we consider that in this type of the studies is a very difficult to obtain the students voluntary participation given the high level of commitment both in time and in behaviour.

Comment 2. Is there any data comparison between different universities in this study? If yes, please explain whether the trend is consistent?

Answer: Thank you for your comment and we regret to say that based on our small sample size we did not perform. Really, there was only one university and the participants were from at 3 different student’s residences

Comment 3. This study should have the analysis data in gender differences. However, the effective sample size is only 47. Is it impossible to conduct gender analysis? Supplementary explanation is suggested.

Answer: Thank you for the comment. We believe that the sample was too small to split it further by sex since the statistical power would have been decreased largely and risking loss of statistical potential.

Comment 4. In order for readers to have a better understanding of the content of the food frequency questionnaires (FFQ) in this study, it is recommended that the complete information of the FFQ be showed as appendix.

Answer: Thank you. The questionnaire (FFQ) has been added as an appendix.

Comment 5. Table 1 and Table 2 are very messy, and there are many abnormal word breaks, which are difficult to read, please adjust.

Answer: Thank you for your valuable comments. We agree with you and the tables have been modified accordingly. We believe that they are now more readable and easy to follow.

Comment 6. Citations listed in this manuscript do not conform to the format of this journal. In addition, links such as DOI or [Google Scholar] [CrossRef] [PubMed] of each article are not listed. Please correct and add relevant information.

Answer: Thank you for your comment. Citations have been modified to conform with the format of the journal  
